# Social Box: Influence of a New Housing System on the Social Interactions of Stallions When Driven in Pairs

**DOI:** 10.3390/ani12091077

**Published:** 2022-04-21

**Authors:** Annik Imogen Gmel, Anja Zollinger, Christa Wyss, Iris Bachmann, Sabrina Briefer Freymond

**Affiliations:** Agroscope, Swiss National Stud Farm, Les Longs Prés, CH-1580 Avenches, Switzerland; annik.gmel@agroscope.admin.ch (A.I.G.); anja.zollinger@agroscope.admin.ch (A.Z.); christa.wyss@agroscope.admin.ch (C.W.); iris.bachmann@agroscope.admin.ch (I.B.)

**Keywords:** stallions, driving, individual housing, social interactions, welfare

## Abstract

**Simple Summary:**

In order to improve the housing conditions of stallions in individual boxes, we tested a so-called “social box” allowing increased physical contact between neighbouring horses. This study aimed at investigating whether housing stallions in social boxes potentially changes their behaviour during carriage driving. We hypothesised that the stay in social boxes would decrease the number of unwanted social interactions when driven in pairs. Eight Franches-Montagnes breeding stallions were observed when driven in pairs with a “neutral” stallion housed in a so-called “conventional box”, strongly limiting physical contact. They were driven on a standardised route over the course of four days before, during, and after being housed in a social box. The behaviours of the pairs and the interventions of the groom and the driver during the test drives were assessed live and using video recordings. The results showed that the stallions performed more social interactions during the driving test before being housed in social boxes and that these interactions decreased over the four days. This suggests that being housed in social boxes decreased the social behaviour of stallions while driven in pairs. Another important factor in reducing unwanted social interactions of stallions during carriage driving appears to be the consistency of the driver and the groom in their demands to teach the stallions that social interactions are unwanted while being driven in pairs. Other effects, such as habituation to the test conditions and the pairing, could not be assessed here and represent a limitation of our study.

**Abstract:**

In order to improve the housing conditions of stallions in individual boxes, we tested a so-called “social box” allowing increased physical contact between neighbouring horses. This study investigated whether housing stallions in social boxes changes the number of social interactions during carriage driving. We hypothesised that the stay in social boxes would decrease the number of unwanted social interactions between stallions when driven in pairs. Eight Franches-Montagnes breeding stallions were observed when driven in pairs with a “neutral” stallion housed in a so-called “conventional box”, strongly limiting physical contact. They were driven on a standardised route over the course of four days before, during, and after being housed in social boxes. The type and frequency of behaviours of the pairs and the interventions of the groom and the driver during the test drives were assessed live and using video recordings. Results from linear mixed-effect models show that unwanted social interactions decreased during and after the stallions were housed in the social box (*p* < 0.001). Stallions’ interactions also decreased over the four days (*p* < 0.01), suggesting a habituation to the test conditions by learning not to interact, or by subtly settling dominance. The social box tended to decrease unwanted social behaviours of stallions driven in pairs and could therefore be used as an environmental enrichment for horses.

## 1. Introduction

Most wild or feral living equids are gregarious, foraging animals living in harem or bachelor bands [1,2], and the domestic horse, *Equus caballus*, has the same ethological need for social contacts with conspecifics [3,4]. However, current housing systems that keep horses in individual stabling tend to strongly limit natural behaviour patterns, such as foraging, locomotor behaviour, and social interactions [5], leading to an increase of stereotypic behaviour in horses housed in sub-optimal conditions [6].

While there are some indications that group housing would potentially improve animal welfare, and despite it being more cost effective [7], a large proportion of horses, and stallions especially, are still mainly kept in individual housing systems. This is partly because many owners perceiving group housing as a potential risk for injuries due to negative interactions between horses, and between stallions in particular [4,8,9]. However, under natural conditions, young wild and feral stallions live together in a bachelor band before establishing their own harem band [8]. Negative social behaviours, such as fights, threats, and agonistic behaviours (avoidance and submissive behaviour), as well as positive social behaviours, such as play fighting and allogrooming, were all observed in stallion bands [9,10,11,12]. Under natural conditions, real fights and injuries due to social interactions are rare, and mostly avoided through ritualised behaviours, settling dominance between horses [9,11,13]. Social contacts are actually essential and play a great role in reducing conflicts and help building a stable hierarchy among the group [14]. Consequently, keeping stallions together has been shown to be possible at least on pasture: the stallions showed mainly ritual and agonistic interactions during the period of integration into the group, and affiliative behaviours increased over time whilst the amount of agonistic interactions decreased [9]. Nonetheless, it is not always possible to keep horses in a group on pasture, for various other practical reasons, such as a lack of space in countries with limited agrarian surfaces, or conversely, a considerable risk of obesity and laminitis due to high-energy grassland pastures. Therefore, adaptations of individual housing systems appear to represent a reasonable compromise between practical constraints and an increase of the possibility for social tactile interactions.

Allowing horses to have more social interactions in their stables may also have positive consequences on their behaviour during training. Previous studies reported increased aggressiveness between horses after social deprivation, especially during ridden training sessions [13,15,16], which could be explained by social frustration and an impairment in social skills induced by limited social contacts between horses [4,17,18]. One can assume that negative social interactions are of particular concern to equestrian disciplines using several horses simultaneously, such as carriage driving. However, to our knowledge, no studies have looked at the potential interactions between horses during carriage driving so far. This equestrian discipline requires a driver (holding the reins), one or more grooms holding the horses while the driver gets on or off the carriage, and standing on the carriage behind the driver during the drive, as well as one, two, or four horses [19]. In practice, for driving, pairs are often chosen according to affinity. However, interactions between horses while driving can be dangerous for both horses and drivers, which is why horses are especially trained not to interact while they are hitched to a carriage in pairs. This is to ensure that the horses are attentive to the driver’s and the groom’s indications, and not to the other horse. Indeed, the forcibly shared personal space (<1 m) could provoke the horses to seek interactions, in order to settle inter-equine conflicts or to play, and ignore human instructions, which might lead to accidents. One essential aim of the driver and grooms is therefore to decrease the number of interactions between horses. This factor is the responsibility of the driver and groom to remain consistent in their demands, by punishing interactions right after they occur, and/or rewarding the absence of interactions, according to learning theory [20]. However, despite their training, interactions between horses may still happen. Another solution could be to decrease the desire for interactions between driven horses by allowing more social interactions in the stables.

With the aim to test a new type of individual housing system, Agroscope evaluated the effect on the behaviour and injury prevalence of a so called “social box” for horses, which allows increased physical contact between the neighbouring stallions by passing their head, neck and/or limbs through vertical bars spaced at 30 cm [21]. The aim of the present study was to investigate whether social interactions in stallions driven in pairs depended on the housing conditions they were living in, i.e., whether the social box potentially might change the social behaviour of Franches-Montagnes stallions when driven in pairs. For the purpose of this study, we built a strong behavioural methodology specifically to measure social interactions during driving sessions. We hypothesise that interactions during the driving exercises would decrease between stallions kept in a social box, due to the possibility for more social interactions in the stables and therefore less frustration-related aggressive behaviours during forced proximity with another stallion at work.

## 2. Materials and Methods

This study was carried out from October 2014 to February 2015 at the Swiss National Stud Farm (“SNSF”) in Avenches, Switzerland, in full compliance with national rules and regulations; under the permit number VD 2810.a.

### 2.1. Animals

For this study, 10 Franches-Montagnes breeding stallions owned by the SNSF were used (mean age ± SD = 8.60 ± 3.17; range 4–14 years). The horses were housed in individual boxes on straw litter and were fed hay and concentrate three times per day (7:00 a.m., 11:00 a.m., 16:00 p.m.). Water was available ad libitum. Horses were exercised 5–6 times a week. During their initial training at 3 years old, all the stallions were ridden and driven to pass the Franches-Montagnes stallion-licensing test. However, after initial training, not all horses participating in this study were still driven regularly. Following legal requirements in Switzerland, all stallions had been socialised in groups after weaning until they arrived as three-year-olds at the SNSF, but extensive tactile contact was limited thereafter due to current housing conditions in individual boxes.

### 2.2. Experimental Housing Facility

All the stables at the SNSF consisted of two rows of four (9.3 m^2^) boxes facing each other and separated by a corridor of 3 m in width. In the so called “conventional box”, the partition between two boxes consisted of a lower solid wooden part (1.40 m high) and an upper part (another 1.15 m high) with vertical metal bars spaced at 5 cm, allowing visual and olfactory contact but strongly limiting tactile contact.

In the so called “social box”, the partition between two boxes consisted of one part with vertical metal bars (from the ground to a height of 2.55 m) spaced at 30 cm allowing the horses to pass their head, neck, and legs to the adjacent box. The second part of the partition was solid, allowing the horses to visually isolate themselves from the neighbouring horse if they wanted to [21]. Each stallion had direct contact with only one neighbour. Each set of two consecutive social boxes was then separated from the next set of social boxes by an opaque partition.

### 2.3. Experimental Design

This investigation was part of a large study investigating physical and behavioural welfare parameters in breeding stallions when housed in individual boxes allowing physical contact [21]. The tested stallions remained for 4 weeks in a conventional box next to their newly assigned neighbour, then spent 4 weeks in the social box, and another 8 weeks in a conventional box with the same neighbour.

To assess whether the stay in a social box was influencing the social behaviour of the stallions during training, eight test stallions were observed driven in pairs with alternatively one of two stallions “neutral” to the treatment. The neutral stallions had never lived in a social box and were housed in conventional boxes in an adjacent barn, with visual, olfactory, and strongly limited tactile contact with other horses (not included in the study). The driven pairs were observed on four days in a row (during four distinct test drives) the week before the test stallions were integrated into a social box (before), the fourth week of housing in the social box (during), and the eighth week after returning from the stay in the social box to a conventional box (after), (Figure 1). Each driven pair acted as its own control. There were only four social boxes in total, so that the eight horses were randomly divided into two groups of four (group A and B), but the exact same timeline was followed in both groups.

### 2.4. Test Drive Procedure

The test drive was composed of three distinct parts: the hitching (putting the stallions in front of the carriage and getting ready for the drive), the actual driving sequence, and the unhitching (releasing the stallions from the carriage) (Figure 1). The route was established beforehand and remained standardised for all drives during the study: the total route included one stop of 10 s, five stops of 30 s, two stops of two minutes and one stop of five minutes. The pace was mostly walk and some trot (no canter). Mean time ± SD per test drive was 38.90 ± 3.03 min.

Each test stallion was driven twice with each neutral stallion, except under specific circumstances such as an unexpected lameness or colic (horse 2, horse 3, and horse 7, see Table 1). As the neutral horses performed two test drives, the order of the test horse was defined so that each test horse was hitched once on the first, and once the second drive of each neutral horse. The stallions were hitched with an English collar to a marathon cart. The bridle was equipped with a straight bar elbow bit and half-cup blinkers. Both driver and groom were equine professionals. The driver was not a collaborator of the SNSF, did not know the stallions beforehand, and was blind to the treatment. The driver was instructed to act consistently over the course of the study, i.e., the interventions (use of rein, whip, or voice) were supposed to be adequate to the intensity of the interactions between the stallions. However, he was instructed not to specifically anticipate and hinder interactions before they occurred. During hitching, the groom had to remain in front of the pair until the start of the driving sequence. The groom was also instructed not to hinder interactions beforehand, i.e., not holding the stallions’ reins during hitching unless the stallions were moving the carriage before the beginning of the driving session.

From the driver’s perspective, the test stallion was always hitched on the left and the neutral stallion on the right-hand side. The driver and the observer were the same for all test drives, while the groom was the same for all test weeks except for Group B “during”. The observer was unfamiliar to the stallions.

### 2.5. Behavioural Observations

The behaviour of the pair, the groom, and the driver was filmed with a GoPro Hero 3 during the entire test drive (except two videos, which were incomplete). The camera was placed on the head of the observer. The hitching and unhitching sequences were filmed standing in front of the carriage, while the driving sequence was filmed by the observer on the carriage, sitting next to the driver. In total, 88 test drives were completed, of which 86 were analysed from video recordings (Table 1). All the behavioural observations were recorded by the same observer using The Observer XT software v.11.5, Noldus Information Technology [22]. Behaviours were recorded as occurrences (“Point Events”). Events longer than one second were counted as multiple events, i.e., one “new” occurrence for each additional second.

#### 2.5.1. Human Interventions

The behaviour of the groom and the driver (human interventions) was first assessed live during the test drive with a portable recording device (Workabout Pro^3^ handheld computer) using the Pocket Observer software [22]. All videos of the test drives were reassessed over the computer in the week following the test drive.

Two types of interventions were counted for the groom, and three for the driver (Table 2). Voice presence was considered a vocal signal directed at both stallions simultaneously. All other driver interventions could be attributed towards a specific stallion. The behaviour of the humans was only assessed if directly related to a social interaction between the stallions, i.e., directly following a stallion interaction. Actions directly relevant for hitching and driving (such as pulling on the reins to turn or stop) and interventions of the groom and driver to stop unwanted behaviour irrelevant to social interactions were thus not assessed.

#### 2.5.2. Interactions between Stallions

The behaviour of the stallions (stallion interactions) was assessed in the weeks following the video analysis of the human interventions using video recordings only. For the video analysis, the order of visualisation was semi-randomised, so as not to watch the same test stallion twice consecutively. The behaviour of the stallions was classified into six occurrences (Table 3). The observer only assessed the behaviour of the stallions directly related to a social interaction, defined as one stallion establishing at least visual contact with the other. Visual contact was the least intensive interaction that could be recognised by the observer. Actions with no obvious social purpose, such as gnawing the driving material, were not assessed. The number of stallion’s interactions was then compared to the number of human interventions.

### 2.6. Statistical Analysis

#### 2.6.1. Descriptive Statistics

Statistical analysis was performed with R software v.4.4.1 [23]. Simple descriptive statistics were computed with the psych package [24]. In a first step, descriptive statistics of the human interventions, i.e., groom and driver, and of the behaviour of the horses were presented. Considering the fact that certain human interventions and horse behaviours were very rare (<5% of all interventions or interactions), the interventions and behaviours were regrouped as follows:-Human interventions were regrouped by person: groom total (“GT”) consisted of the sum of the interventions groom signal (“GS”) and groom presence (“GP”), while driver total (“DT”) consisted of the sum of interventions driver reins (“DR”), driver whip (“DW”), and driver voice (“DV”).-The horse behaviours were regrouped by situation (hitching, standing, moving). Backing up while hitching (“BH”) was regrouped with approach while hitching (“AH”) as hitching total (“HT”). Standing total (“ST”) encompassed approaches while standing (“AS”) and rears while standing (“RS”). Movement total (“MT”) consisted of approaches while moving (“AM”) and rears while moving (“RM”).

All the horse interactions were then regrouped into one category named total interaction (“TI”) and used as the explanatory variable in the analyses. Furthermore, the total human interventions and stallion interactions were compared using Pearson’s correlations.

#### 2.6.2. Statistical Modelling

In order to assess the effect of the housing system (social or conventional boxes), we investigated the effect of the housing system (“Treatment”) on the total number of interactions (“TI”) during the test drive using linear mixed-effect models (LMM; lmer function, lme4 library, [25] in the R software v.4.4.1 [23]). With the assumption that one stallion initiating an interaction should have an influence on the reaction of the other stallion, the sum of interactions of the different pairs (*N* = 16) during the test drive (TI) was included in the model as response variable. The type of housing system (Treatment, “before”, “during”, or “after” being housed in social box), the four different days during which the pairs were observed (four distinct test drives, in each treatment) (“Day”), and whether the neutral stallion was on his first or second test drive (“Drive”) were included as fixed factors. In order to control for repeated measurements of the same pair of stallions, this factor was included as random factor. For all models, the residuals were checked graphically for normal distribution and homoscedasticity (simulate Residuals function, package DHARMa, [26]. For the LMMs, *p*-values (PBmodcomp function, package pbkrtest [27] were calculated using parametric bootstrap methods (1000 bootstrap samples)). Models were fitted with maximum likelihood. The *p*-values calculated with parametric bootstrap tests give the fraction of simulated likelihood ratio test statistic values (LRT) that are larger than or equal to the observed LRT value. This test is more adequate than the raw LRT test because it does not rely on a large-sample asymptotic analysis and correctly takes the random-effects structure into account [27]. *p*-values were extracted by comparing the two models with and without each term, both fitted with the maximum likelihood method (ML), using a likelihood ratio test. The results are presented after model simplification. When a significant interaction effect was found, further two-by-two comparisons were performed using either LMMs or applying a Tukey correction (function glht, package multcomp in R, Multiple comparisons of means). The significance level was set at alpha = 0.05. Moreover, because of the small sample size, we carried out a power analysis for the effect of Day and for the effect of Treatment on Total interactions in order to calculate if the power of our analyses was large enough (pwr.f2 function, pwr library; in R v4.4.1).

#### 2.6.3. Observer Reliability (Intra-Reliability of the Observer)

Intra-observer reliability was calculated for human interventions (live observations vs. first video assessment, first vs. second video assessment) and horse interactions (first vs. second video assessment). For the human interventions, the concordance between live and video observations was evaluated with Cohen’s kappa of the number of interventions on the 86 available test drives to ensure that the video perspective gave the same information as observing the equivalent situation directly. In addition, the intra-rater reliability of video observations was calculated by comparing 32 videos assessed twice. Two videos per tested horse-pair were selected, i.e., 4 videos per tested horse, 16 per control horse, with each horse-pair being observed twice and covering the three treatment situations (at least one test drive for each treatment).

After an initial analysis of the horse behaviours on all videos, 32 videos were also reassessed (four videos per tested horse, 16 per control horse, with each horse-pair being observed twice and at least one test drive for each treatment) to evaluate the intra-rater reliability, similarly to the human interventions. For the video analyses, the order of visualisation was semi-randomised, so as not to watch the same test stallion or treatment twice consecutively. Cohen’s Kappa (κ) [28] was calculated by the Observer XT software [22] for the counted observations of horse and human behaviour for each separate behaviour.

## 3. Results

### 3.1. Behavioural Observations

#### 3.1.1. Descriptive Statistics

##### Stallion Interactions

A mean of 54.87 ± 1.68 interactions in total (TI) were observed per drive, representing approximately 1.5 interactions per minute (Table 4). Across situations, 78.25% of TI were initiated when the horses were not moving (i.e., while standing (AS), 52.51% or hitching (TH), 25.74%), and approaches were the most common behaviour between stallions (25.70% while hitching (AH), 52.12% while standing (AS) and 21.47% during movement (AM)) (Table 4). Some behaviours were specific to an individual, e.g., backs while hitching (BH) was only exhibited by stallion #4. Other types of specific behaviours were seen in several, but not all individuals (rears while standing (RS) in stallion #1, #8, and #9; rears in movement (RM) in stallion #1, #3, #8, #9, and #10, see Appendix A for details). All the behaviours of the stallions are summarised in Appendix A.

##### Human Interventions

Regarding human interventions, 37% could be attributed to the groom and 63% to the driver (Table 5). GP and DV were artificially inflated, as they count twice (for both horses of the pair) while the intervention actually occurs once for the pair. The driver used the reins as the main intervention aid (91% of DT), followed by voice (7% of DT) and whip (1% of DT). Groom presence hardly ever occurred (0.6% of GT) and so groom signals (GS) explained most of the groom interventions. All the human interventions are summarised in Table 5.

### 3.2. Model Results

There was an effect of the statistical interaction between Treatment (before, during or after being housed in social boxes) and Day (Day1 to Day4) on total interactions (TI) (LMM: interaction effect between Treatment and Day on TI, *p* = 0.003, see Appendix A for more details). Power analysis conducted on the results of this model revealed that this model had a power of 100%. Post-hoc comparisons showed that TI differed between Treatment for Day1 (LMM: effect of Treatment on TI, *p* = 0002; Figure 2). Power analysis conducted on the results of this model revealed that this model had a power of *p* > 0.99. During Day1, there was significantly more TI before being housed in social boxes than during and after (Treatment before–during, Multiple comparisons of means Tukey: Z = −4.78, *p* < 0.001; Treatment before–after, Multiple comparisons of means Tukey: Z = 3.65, *p* = 0.0007; Figure 2). However, there was no difference in TI during and after being housed in social boxes (Treatment during and after, Multiple comparisons of means Tukey: Z = −1.37, *p* = 0.36; Figure 2). There was no other effect of Treatment on TI during the other days (LMMs: *p* > 0.77). Moreover, before being housed in social boxes, Day had an effect on TI (LMM: effect of Day on TI, *p* = 0.001, Figure 3). Before being housed in social boxes, there was significantly more TI during Day1 than during Day2, Day3 and Day4 (Day1–Day2, Multiple comparisons of means Tukey: Z = −3.707, *p* < 0.01; Day1–Day3, Multiple comparisons of means Tukey: Z = −3.819, *p* < 0.001, Day1–Day4, Multiple comparisons of means Tukey: Z = −5.467, *p* < 0.01; Figure 3). Power analysis conducted on the results of this model revealed that this model had a power of 100%. However, there was no difference in TI during Day2 and Day3, Day3 and Day4 and Day2 and Day4 (Day2–Day3, Day3–Day4, Day2–Day4, Multiple comparisons of means Tukey: *p* > 0.51; Figure 3). There was also no effect of Day on TI during and after being housed in the social box (LMM: *p* > 0.07). Finally, whether the neutral horse was on its first or second test drive (“Drive”) did not have any effect on TI (LMM: effect of Drive on TI, *p* = 0.12).

### 3.3. Reliability Statistics

In the comparison of driver observations live vs. video (N = 86 comparisons), there was substantial intra-observer reliability, but with a very large range (κ = 0.72, −0.14 < κ < 0.94). A subsampling of 32 videos (two per pair) showed a larger mean κ = 0.75 (0.58 < κ < 0.91) between live and first video analysis, a mean κ = 0.80 for comparisons of videos only (0.59 < κ < 0.96), and a mean κ = 0.73 for comparisons between the live results and the second video analysis (0.57 < κ < 0.90).

For the reliability of the observations of horse interactions, two videos per pair (32 videos in total) were assessed twice. The intra-observer reliability between the first and second video analysis was substantial (κ = 0.75, 0.47 < κ < 0.92).

## 4. Discussion

The results of this study showed that the social box offering opportunities for more physical contact between stallions did not increase their interactions while driven in pairs. This result is consistent with studies testing the influence of the housing system providing social contact between horses on their working ability [13,15,16]. However, our results showed that the horses performed more social interactions during the driving test before being housed in the social box and that these interactions were decreasing over the first few days. Moreover, our aim was also to establish a methodology to measure social interactions during driving sessions. The substantial intra-observer reliability of the human interventions between live observations and observations through video recordings suggests that the GoPro video camera fixed on the head of the observer gave her the same point of view as during live observations. The intra-observer reliability of the horse interactions confirms that they could easily be recorded.

Single stable housing has been demonstrated multiple times to be detrimental to the welfare of horses [6]. This restrictive housing system offers poor opportunities for the horses to fulfil species-specific behaviours, i.e., social interactions and locomotor behaviour, and results in animals staying in a frustration related stress state. This housing-related welfare impairment can lead to abnormal stereotypical behaviour, excessive aggressiveness towards humans [29], unresponsiveness towards their environment and an increase in alert posture [6]. Moreover, it might also negatively affect the training of horses. Previously performed studies on young horses showed that the social environment to which horses are exposed affects learning abilities: young horses stabled in groups were easier to train, less stressed, and demonstrated less aggressive interactions towards humans than single stabled horses [3,15,16]. Concurrently, horses kept in single housing systems performed more aggressive social interactions such as biting in compensation [16]. The need for social interactions might be fulfilled for stallions staying in the social box, replacing the conventional box, and therefore we expected to have a positive effect of the social box by finding less unwanted social interactions when driven in pairs.

Our results showed clearly that the horses performed more social interactions during the first day of the test drives, before being housed in social boxes. Furthermore, the first day generally showed more interactions, and these interactions were decreasing over the four days, especially before being housed in the social box, at the start of the study. As an explanation, we suggest that the decrease in interactions between the first and the following days may indicate a trained response of the stallions, following the consistent demands from the groom and the driver to stop interacting. In terms of learning theory, the driver and the groom added aversive stimuli using the rein, whip, voice, and hand signals in order to decrease the frequency of undesired behaviours, such as social interactions during work in pairs [30]. All stimuli were used in a fairly consistent manner: hand signals and pulling on the reins were most frequent, and of low intensity, while the whip was only used when the horse did not react to the reins or the voice. Such interventions are called positive punishment, and to be effective in learning, the punishing stimulus needs to be applied simultaneously with the undesired behaviour [30]. However, there may be several additional effects concurrent to the effect of the housing system, which could not be addressed in this study and are therefore to be considered as limitation of the study. The decrease between the first and the following days may indicate a certain habituation to the driving partner in the pair by a settling of dominance between the pairs too subtle to be recognised by the observer. We currently cannot disentangle these potential effects from the positive effect attributed to the housing system.

As to our knowledge, so far, no studies have looked at the interactions between horses driven in pairs, and one of our aims was also to validate our methodology for measuring social interactions during driving sessions. We limited the ethogram to behaviours that were obvious to the observer, with the lowest intensity of interaction consisting in one horse turning his head towards the other (“approach”). It is possible that more subtle clues between the two stallions might have been missed and were not recorded for the analysis. However, the substantial intra-observer reliability shows that the behaviours we recorded could be reliably assessed from videos and indicate that the methodology is sound for further research. Furthermore, the intra-observer reliability for the human interventions between live and video observations suggests that either direct observations or assessing on video recordings would be sufficient to record the total interactions between horses driven in pairs. Future studies could use our ethogram as a basis to study the social interactions during driving sessions. For example, some factors, e.g., the friendliness between horses harnessed or individual horses’personality, could influence the number of interactions during the drive, warranting further investigations. Moreover, this ethogram could be used to assess in what kind of situations interactions are most likely to occur, and when they might escalate, resulting in an increased risk of accidents. In our study, most interactions (78.25% of TI) were initiated when the horses were not moving. Therefore, it would be important to train the horses to not interact in these situations. In practice, this would mean planning more stops in the training sessions, where the horses have to stand still, in order to correctly punish the horses to learn not to interact and of course reward them for desired behaviour. The most visibly impressive and potentially dangerous interactions occurred when a stallion reared, either while standing (RS) or even in movement (RM). However, these behaviours were extremely rare. Only one horse backed into the cart while he was being hitched next to the other stallion (BH). This behaviour is dangerous, as the horse may injure himself against the cart, and is also disturbing to the neighbouring horse, but seemed specific to this one individual. Overall, extreme behaviours were rare, and no injuries or accidents occurred.

This article is but a first approach to the complex interactions of horses driven in pairs in response to different housing conditions. It shows several limitations, especially the small number of horses and the limited, though reliable, ethogram, which might have missed more subtle social clues between stallions. No information on the dominance between pairs was established beforehand, which could potentially be a relevant indicator for the subsequent number of interactions between horses. Furthermore, due to the use of neutral stallions in the pair, it is currently unknown whether the stay in the social box between horses driven together would decrease or increase the unwanted interactions during drives even further.

## 5. Conclusions

Our results indicate that being housed in social boxes seems to decrease the number of social interactions between Franches-Montagnes stallions when driven in pairs. Further studies should investigate if factors, such as the affinity between harnessed horses, could influence the number of interactions and whether horses staying together in social boxes would show even fewer interactions when driven together.

## Figures and Tables

**Figure 1 animals-12-01077-f001:**
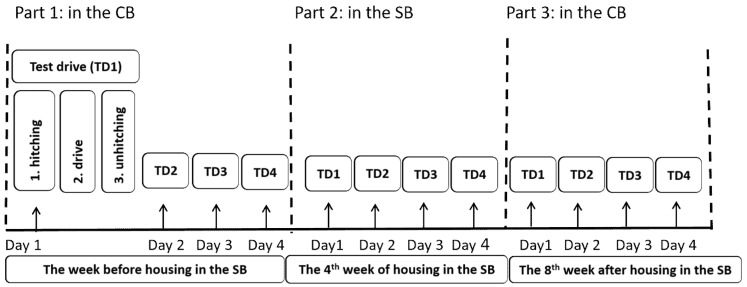
Experimental procedure. The black dotted line indicate the housing system of the stallions at the time during which they were observed (before, during and after being housed in social box). The black arrows indicate the different days during which the drive tests were performed. “CB” is the abbreviation for conventional box, “SB” for social box.

**Figure 2 animals-12-01077-f002:**
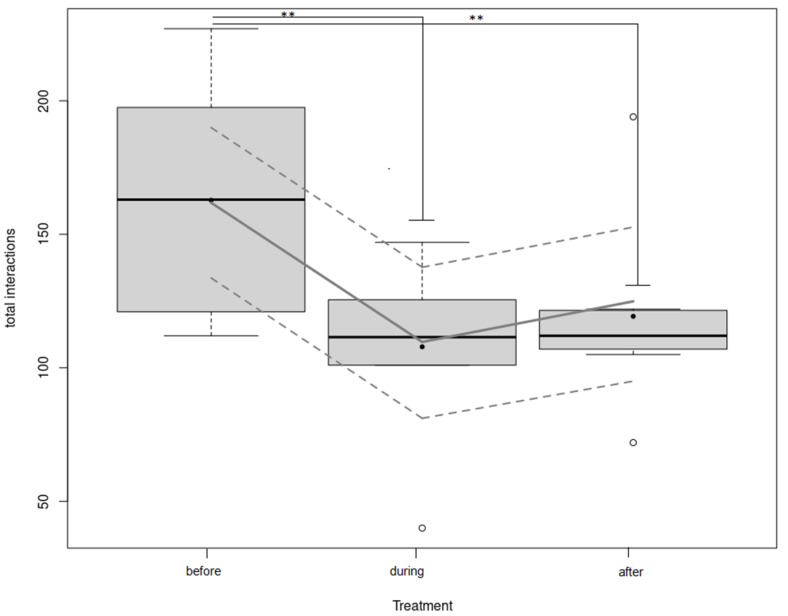
“Total interactions during Day1”. Boxplot of the total interactions recorded in the different treatments (before, during and after being housed in social box) during the first day. The horizontal line shows the median, the box extends from the lower to the upper quartile, and the whiskers to 1.5 × the interquartile range above the upper quartile or below the lower quartile. The black dots indicate the means. Significant differences between treatment are indicated as ** *p* ≤ 0.001. The lines show the model estimates (middle line) and 95% confidence intervals (upper and lower line).

**Figure 3 animals-12-01077-f003:**
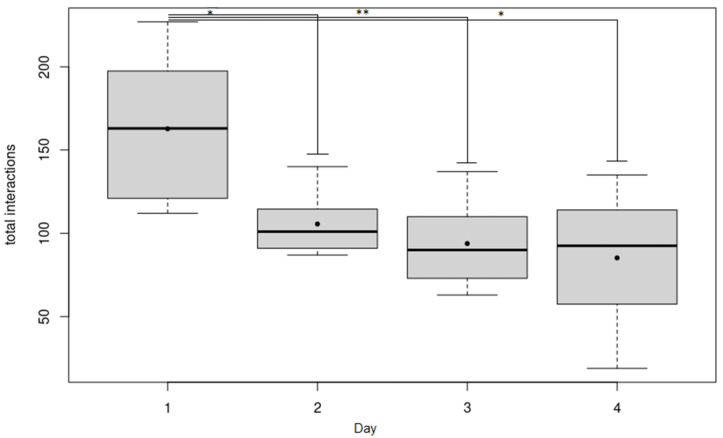
“Total interactions before being housed in social box”. Boxplot of the total interactions recorded during the different days, before being housed in the social box. The horizontal line shows the median, the box extends from the lower to the upper quartile, and the whiskers to 1.5 × the interquartile range above the upper quartile or below the lower quartile. The black dots indicate the mean. Significant differences between treatments are indicated as * *p* ≤ 0.05, ** *p* ≤ 0.001.

**Table 1 animals-12-01077-t001:** Effective pairing of the test (H1–H8) and neutral (N1–N2) horses before, during and after the stay in the social box. Both neutral horses were driven twice per day, so that the first and second drive of the day were also recorded (“drive”). The numbers in bold indicate the days of data collection. Endash (“-“) alone, indicate that the test stallion was driven only once because of specific circumstances such as colic or lameness.

Day	Drive	Before		During		After	
		Group A	Group B	Group A	Group B	Group A	Group B
**1**	1	H2-N1/H8-N2	H4-N1/H5-N2	H2-N1/H8-N2	H1-N1/H4-N1	H2-N1/H8-N2	H4-N1/H5-N2
	2	H6-N1/H3-N2	H7-N1/H1-N2	H6-N1/H3-N2	H7-N2/H5-N2	H6-N2/H3-N2	–/H1-N2
**2**	1	H3-N1/H6-N2	H7-N1/H1-N2	–/H6-N2	H4-N1/H5-N2	H3-N1/H6-N2	–/H1-N2
	2	–/H8-N2	H5-N1/H4-N2	H2-N1/H8-N2	H7-N1/H1-N2	H2-N1/H8-N2	H5-N1/H4-N2
**3**	1	H8-N1/H2-N2	H5-N1/H4-N2	H8-N1/H2-N2	H7-N1/H1-N2	H8-N1/H2-N2	H5-N1/H4-N2
	2	H3-N1/H6-N2	H1-N1/H7-N2	–/H6-N2	H5-N1/H4-N2	H3-N1/H6-N2	H1-N1/–
**4**	1	H6-N1/H3-N2	H1-N1/H7-N2	H6-N1/–	H5-N1/H4-N2	H6-N1/H3-N2	H1-N1/–
	2	H8-N1/H2-N2	H4-N1/H5-N2	H8-N1/H2-N2	H1-N1/H7-N2	H2-N1/H8-N2	H4-N1/H5-N2

**Table 2 animals-12-01077-t002:** Human interventions respective of the groom and driver reacting to social interactions between stallions, highlighted in bold, assessed during the drive test.

Behaviour of the Humans	Description of the Behaviour
**groom signal (GS)**	The groom moves towards the horse and/or touches the horse (or the bridle) to disrupt an interaction between the horses
**groom presence (GP)**	The groom is present at the head of the horses during the drive to deescalate a dangerous situation as perceived by the groom and driver
**driver reins (DR)**	Driver pulls rein on the horse in reaction to a social interaction
**driver whip (DW)**	Driver uses the whip on the horse in reaction to a social interaction
**driver voice (DV)**	Driver uses his voice to hinder a mutual social interaction

**Table 3 animals-12-01077-t003:** Descriptions of stallion interactions highlighted in bold assessed during the test drive.

Behaviour of the Horse	Description of the Behaviour
**Approach while hitching (AH)**	The horse aims his head at the other horse (turns his head approximately 30° towards the other horse), with or without tactile contact, during hitching procedures
**Backs while hitching (BH)**	The horse backs into the cart to avoid hitching after eye contact with other horse
**Approach while standing (AS)**	The horse aims his head at the other horse (turns his head approximately 30° towards the other horse), with or without tactile contact, while standing
**Rear while standing (RS)**	The horse rears in the direction of the other horse while standing
**Approach in movement (AM)**	The horse aims his head at the other horse (turns his head approximately 30° towards the other horse), with or without tactile contact, while in movement
**Rear in movement (RM)**	The horse rears in the direction of the other horse while in movement

**Table 4 animals-12-01077-t004:** Descriptive statistics of the number of observed horse interactions between stallions driven in pairs highlighted in bold over the course of the study (Total number of occurrences; mean ± SE; min-max; percentage of total recorded behaviours). The behaviours in italics indicate the behaviours regrouped by type of behaviour. The number of occurrences in italics indicates the number of occurrences for the behaviours regrouped by type of behaviours. The number of occurrences in italics and bold indicates the number of occurrences for the total driving. The number of occurrences in bold indicates the number of occurrences for the total interactions. See Table 3 for the detailed ethogram.

Behaviour of the Horses	Total Number (Over All Drives)	Mean ± SE	Minimum-Maximum	Percentage of Total Recorded Behaviours
**Approach while hitching (AH)**	2426	14.1 ± 0.6	0–39	25.70%
**Backs while hitching (BH)**	3	0.02 ± 0.01	0–2	0.03%
** *Total hitching (TH = AH + BH)* **	*2429*	*14.12 ± 0.6*	*0–39*	*25.74%*
**Approach while Standing (AS)**	4919	28.60 ± 1.20	0–89	52.12%
**Rear while standing (RS)**	37	0.22 ± 0.09	0–10	0.39%
** *Total standing (TS = AS + RS)* **	*4956*	*28.81 ± 1.2*	*0–89*	*52.51%*
**Approach in movement (AM)**	2026	11.78 ± 0.72	0–69	21.47%
**Rear in movement (RM)**	27	0.16 ± 0.05	0–6	0.29%
** *Total movement (TM = AM + RM)* **	*2053*	*11.94 ± 0.72*	*0–69*	*21.75%*
** *Total driving (TD = TS + TM)* **	* **7009** *	* **40.75 ± 1.52** *	* **3–134** *	* **74.26%** *
**Total interactions (TI** **= *TH + TS + TM*** **)**	**9438**	**54.87 ± 1.68**	**8–151**	**100.00%**

**Table 5 animals-12-01077-t005:** Descriptive statistics of the number of observed human interventions to stop stallions driven in pairs from interacting over the course of the study (Total number of occurrences mean ± SE; min-max; percentage of total recorded interventions). The number of occurrences in italics indicates the number of occurrences for the total groom, resp. total driver interventions. The number of occurrences in italics and bold indicates the number of occurrences for the total groom and driver interventions. See Table 2 for the detailed ethogram.

Human Interventions	Total Number (Over All Drives)	Mean ± SE	Minimum-Maximum	Percentage of Total Recorded Behaviours
**Groom Signal (GS)**	2399	13.95 ± 0.61	0–34	37.23%
**Groom Presence (GP)**	14	0.08 ± 0.04	0–4	0.22%
** *Total groom (TG = GS + GP)* **	*2413*	*14.03 ± 0.61*	*0–34*	*37.45%*
**Driver Reins (DR)**	3678	21.38 ± 1.31	0–108	57.08%
**Driver Whip (DW)**	63	0.37 ± 0.08	0–6	0.98%
**Driver Voice (DV)**	290	1.69 ± 0.19	0–13	4.50%
** *Total driver (TD = DR + DW + DV)* **	*4031*	*23.44 ± 1.41*	*0–110*	*62.55%*
**Total human interventions (** **TG + TD** **)**	**6444**	**37.47 ± 1.67**	**0–135**	**100.00%**

## Data Availability

Data are available upon reasonable request to the corresponding author.

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
