# Peer review of "Social Box: Influence of a New Housing System on the Social Interactions of Stallions When Driven in Pairs"

_animals, 2022, doi:10.3390/ani12091077_

Round 1
Reviewer 1 Report
This original research looks at the effect of housing stallions together with the ability for limited social interaction on their agreeableness during work (carriage driving). This is an important study that adds information to improve our knowledge of best management practices for housing stallions.
The introduction is well written presenting relevant information leading up to the objectives and hypotheses. It could benefit from an English grammar check.
Methods: it would be useful to provide a photo of both the conventional box and the social box. Were the neutral stallions familiar to the test stallions? Or had they never met prior to the first drive?
The first paragraph of the discussion sums up your results excellently! In fact the whole discussion is well written. However one question I had was whether the interactions between stallions while hitched would differ (be less?) if they were hitched with a stallion who was their neighbor in the social box. Could it be that having the opportunity to interact outside of “work” would enable the stallions to focus on work better? Having an average of 1.5 interactions/min seems quite high, and even though the horses are responding to training, these interactions are not being eliminated.
Any limitations to the study?
Please check referencing guidelines. I don’t believe the references are in the proper format.
Many of the references are also rather dated. Likely the authors looked for more recent papers, but please ensure that more recent publications have not been excluded.
Some minor comments below:
L149 – were the test drives during Part 3 done after the stallions had been moved back to conventional boxes? Was it the first 4 days after moving back or were they acclimated to their new environment first?
L190 – where was the observer in relation to the driven horses, driver and groom? Oh, I see. Perhaps move the last sentence to immediately after the sentence indicating the camera was placed on the head of the observer to flow better. From sitting beside the driver, how was the observer able to video the driver and the horses at the same time?
L201-208 and 227-230 – these lines describe how you calculated inter-observer reliability, which is repeated in section 2.6.3. Perhaps only need to include in the latter section.
L211-215 – I think I understand what you mean here but the sentence is confusing. Maybe reword to something like “Human interventions were attributed to either the groom or the driver (Table 2).” Further I’m not sure I understand what is meant by these interventions represented <5% of total observations. Perhaps this whole paragraph can be made clearer?
L232-239 – similar to comment above, this is a bit confusing (although better worded than above). I think I understand that some observations were infrequent, thus they were grouped into a larger category for statistical analyses. Perhaps it might be more clear if this was explained in the analysis section.
L304 – Table captions should provide enough information for the reader to understand the table without having to read all the text. Please expand the caption to include information such as the fact that it was looking at stallion interactions during a driving test and how many stallions were observed. Also it would be helpful in Table 4 to perhaps draw horizontal lines to divide the three sections (hitching, standing, movement) or to bold the lines for TH, TS, TM, TI.
L312 – same comments as for Table 4
L315 – I don’t follow the numbering here. Should this not be 3.2? Also the number 1 looks like superscript.
L343 – remove the title from the figure itself and include in the caption. Same for Fig 3
L422 – typo “life” should be “live”
Author Response
Reviewer #1 :
This original research looks at the effect of housing stallions together with the ability for limited social interaction on their agreeableness during work (carriage driving). This is an important study that adds information to improve our knowledge of best management practices for housing stallions.
R: We thank the reviewer for this positive evaluation of our manuscript and for the very useful comments.
The introduction is well written presenting relevant information leading up to the objectives and hypotheses. It could benefit from an English grammar check.
R: We thoroughly revised the introduction to address any language issues (e.g. LL43, 64-65, 80-82, 86-87, 90-91, 312-315)
Methods: it would be useful to provide a photo of both the conventional box and the social box. Were the neutral stallions familiar to the test stallions? Or had they never met prior to the first drive?
R: Photos of the conventional and social box will be presented in the article specific to the social box (Zollinger et al., in preparation). The neutral horses were driven regularly with other stallions, but not the ones from the study. The test stallions were only driven in pairs during the duration of the study. It is possible that they were paired before, but not since starting the study at least 3 weeks before the test drives.
The first paragraph of the discussion sums up your results excellently! In fact the whole discussion is well written. However one question I had was whether the interactions between stallions while hitched would differ (be less?) if they were hitched with a stallion who was their neighbor in the social box. Could it be that having the opportunity to interact outside of “work” would enable the stallions to focus on work better? Having an average of 1.5 interactions/min seems quite high, and even though the horses are responding to training, these interactions are not being eliminated.
R: While we hypothesized that the interactions would decrease with the stay in the social box, from a practical perspective, we could not be sure and wanted to have the “neutral” horses as safeguards during the experiment. The effect of keeping the same pairs of horses in the social box and drive them in pairs is a very interesting question we might investigate in a future experiment. The hypothesis that they would interact less sounds logical and should definitively be tested. The frequency of interactions is high also because the test drive included longs “stops” where the horses interact the most. Under real conditions, the usual drivers avoid long stops to avoid having too much interactions (or they get help from a groom on the ground, standing in front of the horses, preventing interactions). It’s also to notice, that we count longer interactions as several occurrences.
Any limitations to the study?
R: We have added a paragraph on limitations of the study in the discussion L474-482.
Please check referencing guidelines. I don’t believe the references are in the proper format.
R: We revised the format of the references.
Many of the references are also rather dated. Likely the authors looked for more recent papers, but please ensure that more recent publications have not been excluded.
R: We checked the publications and added recent paper in the introduction e.g. L. 47
Some minor comments below:
L149 – were the test drives during Part 3 done after the stallions had been moved back to conventional boxes? Was it the first 4 days after moving back or were they acclimated to their new environment first?
R: For part 3, they were tested 8 weeks after returning to the conventional box, so after they had time to readapt to conventional housing (see line 142-144.)
L190 – where was the observer in relation to the driven horses, driver and groom? Oh, I see. Perhaps move the last sentence to immediately after the sentence indicating the camera was placed on the head of the observer to flow better. From sitting beside the driver, how was the observer able to video the driver and the horses at the same time?
R: Thank you for the suggestion, we moved the sentence earlier in the text. The GoPro Hero 3 has a very wide angle lens, making it possible to see the hands and sometimes the upper body of the driver as well.
L201-208 and 227-230 – these lines describe how you calculated inter-observer reliability, which is repeated in section 2.6.3. Perhaps only need to include in the latter section.
We combined that section with the latter one (section 2.6.3, lines 290-303)
L211-215 – I think I understand what you mean here but the sentence is confusing. Maybe reword to something like “Human interventions were attributed to either the groom or the driver (Table 2).” Further I’m not sure I understand what is meant by these interventions represented <5% of total observations. Perhaps this whole paragraph can be made clearer?
We have moved this part to the analysis section and rephrased it to make it clearer (line 240-254).
L232-239 – similar to comment above, this is a bit confusing (although better worded than above). I think I understand that some observations were infrequent, thus they were grouped into a larger category for statistical analyses. Perhaps it might be more clear if this was explained in the analysis section.
We have moved this part to the analysis section and rephrased it to make it clearer (line 240-254).
L304 – Table captions should provide enough information for the reader to understand the table without having to read all the text. Please expand the caption to include information such as the fact that it was looking at stallion interactions during a driving test and how many stallions were observed. Also it would be helpful in Table 4 to perhaps draw horizontal lines to divide the three sections (hitching, standing, movement) or to bold the lines for TH, TS, TM, TI.
We have adapted the title and also layout of the tables (line 322-325), in the hopes that it is acceptable to the editor.
L312 – same comments as for Table 4
We have adapted the title and also layout of the tables (line 335-339), in the hopes that it is acceptable to the editor.
L315 – I don’t follow the numbering here. Should this not be 3.2? Also the number 1 looks like superscript.
R: We revised the numbering as suggested LL323, 329, 367
L343 – remove the title from the figure itself and include in the caption. Same for Fig 3
R: We revised this Figure L 368 and L 376.
L422 – typo “life” should be “live”
R: We revised the word as suggested L454.
Reviewer 2 Report
This paper compares the effects of housing (“traditional”, allowing very limited social contact with conspecifics, and “social”, allowing a larger degree of social contact with conspecifics) on the behaviour of stallions whilst being driven in pairs. Whilst the sample size was limited to eight stallions, appropriate statistical techniques were used to analyse the dataset and allow for this small sample size.
This is an interesting study as few researchers have focused on the behaviour of horses whilst being driven, and housing that allows increased levels of social interactions is becoming more popular. Whilst this study is generally well communicated, there are some aspects of the methods that require some clarity for its results and interpretations to be understood, as outlined below.
One concern I do have is in the authors’ interpretation of the results, if I have understood the methods correctly (and there is some ambiguity in this section). If the two stallions have been driven together frequently during the study, but day one was their first time together, the results in figure 2 (a reduction in social interactions during and after being housed in the social box) could just be explained by the stallions getting to know each other over time and no longer requiring social interactions to define their relative dominance. A similar explanation could be used for figure 3. Therefore the authors may not actually be testing the effects of a social box in this study; the experimental design means there is no control to compare with (i.e. stallions not housed in a social box, but driven in the same pair over time, were not tested). This at least needs to be considered in the discussion section.
Introduction
This gives a clear rationale for the study, summarising our current understanding of the reasons for and against social housing for stallions.
Methods
Some points need to be clarified for the methods to be understood. Firstly, it is not clear for how long the test stallions were housed in the social boxes. Were they tested for a week in traditional boxes, then housed for four weeks in social boxes, being tested in the fourth week, then housed for a further eight weeks in traditional boxes, being tested on the eighth week? Or was this final testing phase week 8 of the study, meaning it was in the fourth week after they returned to traditional boxes?
Secondly, it is my understanding that all eight stallions were housed in pairs that were consistent throughout the study, i.e. during traditional and social housing. However, for this experiment, they were never driven in these pairs. I understand that you wanted to isolate the behaviour of the “test” horses by driving them in a pair with the “neutral” horses (which had never experienced social housing), but would it have not been more realistic/useful to drive horses in pairs that had experienced social interactions via social stabling, so that they had ample time to interact and therefore secure their social standing with each other outside of driving time? I think this would have been a more useful experiment, comparing these pairs with pairs that had not had this opportunity.
Thirdly, please clarify how frequently the stallions were driven (and with whom) between the test periods. Also please clarify the order of each test horse being driven with each neutral horse (i.e. did these pairings alternate? Is this what you mean by a Latin square design?) and whether they had experienced these pairings prior to the study.
Specific points for clarification:
132-138 – can you clarify for the reader the length of the section containing the 30cm-spaced bars? Did this take up half the partition? Was it at the front of the stable or the back?
211-212 – rephrase – “The behaviour of the groom was classified into two distinct categories, whilst that of the driver was classified into three categories.”
212-213 – rephrase – do you mean that these behaviours in total formed less than 5% of total experimental time?
217-218 – how was it objectively determined whether the groom/driver’s behaviour was directly as a result of stallions’ behavioural interactions? Was a rule used e.g. that it had to occur within 2 seconds of a behavioural interaction?
232-235 – rephrase – not clear what is meant here
241-242 – I’m not sure how you determined that social behaviour was only present when stallions established visual contact – eye contact is not needed for horses to communicate, and would have been difficult when these horses were wearing blinkers. Do you mean something more specific, such as when one horse turned its head at least xxx degrees towards the other stallion?
Table 3 – again a bit more detail – how would you define aiming head towards other stallion? And why backing into cart after eye contact? See previous point – eye contact not required for horses to interact socially
253-254 – clarify whether it was the total no. interactions per minute or some other measure that was compared by correlations
261-262 – do you mean the total no. of incidences of each behaviour during each driving test was used as the outcome variable? Please clarify
264-265 – whether the neutral horse was on his first or second test drive of the day – I’m not clear what is meant here, since I thought each of the test stallions was driven with each neutral stallion twice during a week, so how could the two neutral stallions only be driven twice per day?
Results
This section is clearly presented.
Discussion
To fully evaluate your interpretations of results, I would like to clarify whether a) each pair of stallions had been driven together before the study and b) the order of pairings throughout the four day period. If the two were first driven together on day 1 (either they had never been previously driven together, or not for a long time), you would expect them to show a higher level of social interactions, which then decreased on days 2, 3 or 4 (depending on order of pairings with each neutral horse) as they were then familiar to each other and had ascertained dominance. This could be an alternative explanation to the authors’ explanation that the stallions were being “trained” not to interact by the driver. My explanation is given in lines 413-5 as an alternative, but I feel it deserves a lot more consideration as it is more likely to explain the observed decrease in interactions than “training”. These are, I assume, horses that have considerable experience of being driven; they are likely to have learnt early on in their career that social interactions will result in punishment, so it is unlikely they are learning this for the first time here. If they are being “trained” here, this probably shows that positive punishment is not the most effective training method for younger horses as they have obviously not learnt this earlier in their career.
368-370 – Actually, I think you have shown that housing horses in a social box significantly decreases unwanted social interactions when driven. Surely this is stronger than what you have said – that this housing method does not increase social interactions?
428 – remove the word “validated”; the ethogram has been used effectively, but it has not been validated. Instead, the method of filming has been validated
Author Response
Reviewer # 2 :
Comments and Suggestions for Authors
This paper compares the effects of housing (“traditional”, allowing very limited social contact with conspecifics, and “social”, allowing a larger degree of social contact with conspecifics) on the behaviour of stallions whilst being driven in pairs. Whilst the sample size was limited to eight stallions, appropriate statistical techniques were used to analyse the dataset and allow for this small sample size.
This is an interesting study as few researchers have focused on the behaviour of horses whilst being driven, and housing that allows increased levels of social interactions is becoming more popular. Whilst this study is generally well communicated, there are some aspects of the methods that require some clarity for its results and interpretations to be understood, as outlined below.
One concern I do have is in the authors’ interpretation of the results, if I have understood the methods correctly (and there is some ambiguity in this section). If the two stallions have been driven together frequently during the study, but day one was their first time together, the results in figure 2 (a reduction in social interactions during and after being housed in the social box) could just be explained by the stallions getting to know each other over time and no longer requiring social interactions to define their relative dominance. A similar explanation could be used for figure 3. Therefore the authors may not actually be testing the effects of a social box in this study; the experimental design means there is no control to compare with (i.e. stallions not housed in a social box, but driven in the same pair over time, were not tested). This at least needs to be considered in the discussion section.
R: We have now considered these effects in the discussion, thank you for suggesting it.
Introduction
This gives a clear rationale for the study, summarising our current understanding of the reasons for and against social housing for stallions.
R: Thank you for the assessment
Methods
Some points need to be clarified for the methods to be understood. Firstly, it is not clear for how long the test stallions were housed in the social boxes. Were they tested for a week in traditional boxes, then housed for four weeks in social boxes, being tested in the fourth week, then housed for a further eight weeks in traditional boxes, being tested on the eighth week? Or was this final testing phase week 8 of the study, meaning it was in the fourth week after they returned to traditional boxes?
R: We have changed Figure 1 to clarify the timeline, and added a sentence to clarify:
“The tested stallions remained four weeks in a conventional box next to their newly assigned neighbour, then four weeks in the social box, and another eight weeks in the conventional box with the same neighbour.” (line 142-144)
Secondly, it is my understanding that all eight stallions were housed in pairs that were consistent throughout the study, i.e. during traditional and social housing. However, for this experiment, they were never driven in these pairs. I understand that you wanted to isolate the behaviour of the “test” horses by driving them in a pair with the “neutral” horses (which had never experienced social housing), but would it have not been more realistic/useful to drive horses in pairs that had experienced social interactions via social stabling, so that they had ample time to interact and therefore secure their social standing with each other outside of driving time? I think this would have been a more useful experiment, comparing these pairs with pairs that had not had this opportunity.
R: While we hypothesized that the interactions would decrease with the stay in the social box, from a practical perspective, we could not be sure and wanted to have the “neutral” horses as safeguards during the experiment. The effect of keeping the same pairs of horses in the social box and drive them in pairs is a very interesting question we might investigate in a future experiment. The hypothesis that they would interact less sounds logical and should definitively be tested.
Thirdly, please clarify how frequently the stallions were driven (and with whom) between the test periods. Also please clarify the order of each test horse being driven with each neutral horse (i.e. did these pairings alternate? Is this what you mean by a Latin square design?) and whether they had experienced these pairings prior to the study.
R: The neutral horses were driven regularly with other stallions, but not the ones from the study. The test stallions were only driven in pairs during the duration of the study. It is possible that they were paired before, but not since starting the study at least 3 weeks before the test drives.
Specific points for clarification:
132-138 – can you clarify for the reader the length of the section containing the 30cm-spaced bars? Did this take up half the partition? Was it at the front of the stable or the back?
R: The bars go from the ground to the top (2.55 m high). We rephrased this to be more clear (line 133)
211-212 – rephrase – “The behaviour of the groom was classified into two distinct categories, whilst that of the driver was classified into three categories.”
R: We rephrased this section entirely (line 211, and line 243-246)
212-213 – rephrase – do you mean that these behaviours in total formed less than 5% of total experimental time?
R: We rephrased this section (line 240-253), but all recorded behaviours were “point events” (see line 204-205), so the 5% means 5% of counted behavioural occurrences.
217-218 – how was it objectively determined whether the groom/driver’s behaviour was directly as a result of stallions’ behavioural interactions? Was a rule used e.g. that it had to occur within 2 seconds of a behavioural interaction?
R: The behavior directly followed the interaction of the horse. We have not used a particular time limit.
232-235 – rephrase – not clear what is meant here
R: We have rephrased and moved this section to the reliability analysis section. (line 290-303).
241-242 – I’m not sure how you determined that social behaviour was only present when stallions established visual contact – eye contact is not needed for horses to communicate, and would have been difficult when these horses were wearing blinkers. Do you mean something more specific, such as when one horse turned its head at least xxx degrees towards the other stallion?
R: We had to determine the most subtle interaction that could still be discerned by the observer, which is why we limited the interactions to visual contact (more specifically: turn the head). The horses had to turn their head approximately 30° degrees towards their neighbor. We have added this limitation in the methods (line 229 and table 3) and the discussion (line 447-450) sections. The eye contact was made possible by the blinkers being half-cups, protecting the eye of the horse from an inadvertent stroke with the whip, but otherwise not restricting vision to the side or the front.
Table 3 – again a bit more detail – how would you define aiming head towards other stallion? And why backing into cart after eye contact? See previous point – eye contact not required for horses to interact socially
R: We have added the information that the head was turned at an angle of approximately 30° towards its neighbor (table 3). The behavior “backing into cart” was observed exactly in this way, therefore this is how it was described. We have been able to record no other behaviours than the ones presented reliably.
253-254 – clarify whether it was the total no. interactions per minute or some other measure that was compared by correlations
R: We have removed the correlation section.
261-262 – do you mean the total no. of incidences of each behaviour during each driving test was used as the outcome variable? Please clarify
R: We mean the total number of incidences of all behaviours during each test drive was used as the outcome variable. We have reformulated the paragraph to be more clear (line 253-254).
264-265 – whether the neutral horse was on his first or second test drive of the day – I’m not clear what is meant here, since I thought each of the test stallions was driven with each neutral stallion twice during a week, so how could the two neutral stallions only be driven twice per day?
R: We have adapted the material and methods to explain this better (line 173-175), and adapted the table to be clearer. As only 4 boxes and 2 neutral horses were available, the 8 test horses were separated into two groups and tested in two different weeks.
Results
This section is clearly presented.
R: Thank you
Discussion
To fully evaluate your interpretations of results, I would like to clarify whether a) each pair of stallions had been driven together before the study and b) the order of pairings throughout the four day period. If the two were first driven together on day 1 (either they had never been previously driven together, or not for a long time), you would expect them to show a higher level of social interactions, which then decreased on days 2, 3 or 4 (depending on order of pairings with each neutral horse) as they were then familiar to each other and had ascertained dominance. This could be an alternative explanation to the authors’ explanation that the stallions were being “trained” not to interact by the driver. My explanation is given in lines 413-5 as an alternative, but I feel it deserves a lot more consideration as it is more likely to explain the observed decrease in interactions than “training”. These are, I assume, horses that have considerable experience of being driven; they are likely to have learnt early on in their career that social interactions will result in punishment, so it is unlikely they are learning this for the first time here. If they are being “trained” here, this probably shows that positive punishment is not the most effective training method for younger horses as they have obviously not learnt this earlier in their career.
R: a) The neutral horses were driven regularly with other stallions, but not the ones from the study. The test stallions were only driven in pairs during the duration of the study. It is possible that they were paired before, but not since starting the study at least 3 weeks before the test drives.
- b) We have reformulated and improved the table describing the pairing of the horses, so this should be easier to understand now. However, as the statistical factor “drive” was not significant, the order of the test drive did not seem to have an effect on the outcome.
It is possible that settled dominance may have played a large effect in the reduction of interactions. However, it is not possible with this dataset to disentangle these effects. We have added a limitations section to the discussion to take these effects into account (line 440-443, 449-450).
All the horses in this study have learnt how to be driven at an early age (3 years old), but were not necessarily driven in pairs very often since (except for the neutral stallions), which could mean that the horses had to be “reminded” of the rules of not interacting. Furthermore, the highest amount of interactions were taking place during the times the horses were standing (“stops”), a situation which is not often incorporated into usual training sessions (from our empirical assessment). The usual drivers avoid long stops to avoid having too much interactions (or they get help from a groom on the ground, standing in front of the horses, preventing interactions).
368-370 – Actually, I think you have shown that housing horses in a social box significantly decreases unwanted social interactions when driven. Surely this is stronger than what you have said – that this housing method does not increase social interactions?
R: Due to the diverse additional effects you have also mentioned (habituation, hierarchy settling, training response), we do not want to make a more decided conclusion on this subject.
428 – remove the word “validated”; the ethogram has been used effectively, but it has not been validated. Instead, the method of filming has been validated
R: The word validate has been removed
Reviewer 3 Report
Feedback for Authors
Title: Good
Abstract:
Second to last sentence (lines 35 – 36) needs some more explanation – I did not quite understand how that conclusion (the consistency of the driver being the most important) followed on logically from the previous information.
Introduction:
Lines 50 – 52, sentence needs a small restructure – lack of space seems to suggest stall housing, but the next part of the sentence is talking about grassland pastures- which is presumably not a space problem. I understand what you mean, but it could be expressed more clearly.
Lines 85 – 87, sentence needs a bit of clarification – are the grooms also sitting on the carriage during the drive or is it just the driver?
Aims are clear and testable.
Material and Methods:
Experimental procedure very clearly described
2.5.1 – any data on the inter-rater and inter-modal reliability measures? Oh – I see it is in the Results section – might it be better placed in the Methods section as a validation of the tools used, rather than at the end of the Result section?
Clear description of behaviours observed and measured.
Supplementary Table S1. A reminder of the abbreviated behaviours would have been useful for the reader. Why are the last two horses in their own section? Is it because they displayed so many more interaction behaviours than the other horses?
Results:
3.1.1.2 – if the authors are aware of the (understandable) artificial inflating of some of the human interventions, would it make sense to adjust this? (Lines 305 – 307 and Table 5).
- Correlations (lines 315 – 319). I am not entirely clear on the value of this correlation. It either needs a bit more explanation of what it contributes (whilst of course you cannot discuss it in the Results section), or remove this small section altogether.
- Model results (Lines 321 – 342). With such a small sample size, there is limited value in the inferential statistics. I understand that journals often require this, but with single-subject design (or a low number of subjects) such as yours, visual representation of the results is often more appropriate.
Your graphs are a much stronger indication of the effects of social boxes on interactions. I would like to suggest that you expand on these: with such a lot of difference between horses in the number of interactions that they display, and – as you point out – the horses are their own control, it would be really useful to examine and show how individual horse behaviour changed between the before and during conditions. The continued lower rates in interactions in the after condition is a really nice finding too as it suggests longer term positive (reduced) effect of social housing on interactions in the driving situation.
Discussion:
Line 374 suggest re-wording to “This suggests that…”
Lines 376 – 378, 404 - 407 – I do not think that the correlation really shows this (human behavioural consistency). Your mention of decrease in interactions over time is stronger evidence.
Line 391 suggest re-wording to “Previous studies…”
Line 401 suggest re-wording to “and these interactions decreased…”
Line 416 suggest re-wording “To our knowledge…”
Line 422 “life” should be “live observations”
Line 433 suggest re-wording to “planning more breaks”
Line 434 needs rewording.
Lines 435 – 438 needs rewording and the RM abbreviation is bolded when other abbreviations are not.
Just a thought – and I know it is not possible to collect new data on this, but it could be mentioned as a suggested future study: why not pair the horses driving who are housed together in the social boxes? As well as (or instead of) the neutral horses? Any info on dominance hierarchy between the horses?
Overall comment: this is a very interesting and valuable paper which is overall written very well. I think that you have an opportunity to present some detailed individual change data which may show strong effects of the social housing conditions as the mean data (which is so variable) may hide individual changes in interactions.
I enjoyed reading this, thank you.

Author Response
# Reviewer 3
Feedback for Authors
Title: Good
Abstract:
Second to last sentence (lines 35 36) needs some more explanation I did not quite understand how that conclusion (the consistency of the driver being the most important) followed on logically from the previous information.
R: We have removed this from the abstract, as it was related to a subsection we removed on behalf of another reviewer.
Introduction:
Lines 50 52, sentence needs a small restructure lack of space seems to suggest stall housing, but the next part of the sentence is talking about grassland pastures- which is presumably not a space problem. I understand what you mean, but it could be expressed more clearly.
R: We reformulated and relocated the sentences: “…various practical reasons such as a lack of space in countries with limited agrarian surfaces, or conversely, a considerable risk of obesity and laminitis due to high-energy grassland pastures.” (line 65-67)
Lines 85 87, sentence needs a bit of clarification are the grooms also sitting on the carriage during the drive or is it just the driver?
R: The groom sits or stands behind the driver during the drive, to be ready to intervene near the horse’s head, but especially to balance the movement of the cart during high speed maneuvers. We modified this in the manuscript. (line 80-92)
Aims are clear and testable.
R: Thank you
Material and Methods:
Experimental procedure very clearly described
R: Thank you
2.5.1 any data on the inter-rater and inter-modal reliability measures? Oh I see it is in the Results
section might it be better placed in the Methods section as a validation of the tools used, rather than at the end of the Result section?
R: As the validation is part of the aims of this study, we would rather keep it in the results section.
Clear description of behaviours observed and measured.
R: Thank you
Supplementary Table S1. A reminder of the abbreviated behaviours would have been useful for the reader. Why are the last two horses in their own section? Is it because they displayed so many more interaction behaviours than the other horses?
R: We have adapted the table with a reminder of the abbreviations.
The last two horses represent the neutral stallions, who have performed double the amount of test drives. This is why they are apart from the eight test stallions that were tested in the social box. However, it might still be useful to see that there were also individual differences between the neutral stallions.
Results:
3.1.1.2 if the authors are aware of the (understandable) artificial inflating of some of the human
interventions, would it make sense to adjust this? (Lines 305 307 and Table 5).
R: We have attributed it to both horses, and in a direct comparison between the total number of interventions and interactions this therefore matters.
- Correlations (lines 315 319). I am not entirely clear on the value of this correlation. It either needs a bit more explanation of what it contributes (whilst of course you cannot discuss it in the Results section), or remove this small section altogether.
R: We have removed this section.
1.2 Model results (Lines 321 342). With such a small sample size, there is limited value in the inferential statistics. I understand that journals often require this, but with single-subject design (or a low number of subjects) such as yours, visual representation of the results is often more appropriate.
R: As we prefer to use models, we added power analyses to support our models. L282-285, L344-345, L347-348, L360-361.
Your graphs are a much stronger indication of the effects of social boxes on interactions. I would like to suggest that you expand on these: with such a lot of difference between horses in the number of interactions that they display, and as you point out the horses are their own control, it would be really useful to examine and show how individual horse behaviour changed between the before and during conditions. The continued lower rates in interactions in the after condition is a really nice finding too as it suggests longer term positive (reduced) effect of social housing on interactions in the driving situation.
R: As suggested, we added a Supplementary Material S2, in which boxplot for each different pair are represented, i.e. the total interactions recorded in the different treatments (before, during and after being housed in social box).
Discussion:
Line 374 suggest re- that
R: We reworded it like this (line 390-392):
“For the reliability of the observations of horse interactions, two videos per pair (32 videos in total) were assessed twice. The intra-observer reliability between the first and second video analysis was substantial (κ=0.75, 0.47 < κ < 0.92).”
Lines 376 378, 404 - 407 I do not think that the correlation really shows this (human behavioural consistency). Your mention of decrease in interactions over time is stronger evidence.
R: We have removed the correlations altogether.
Line 391 suggest re-
R: We reworded the sentence this way (line 402-406):
“The substantial intra-observer reliability of the human interventions between live observations and observations through video recordings suggests that the GoPro video camera fixed on the head of the observer gave her the same point of view as during live observations. The substantial intra-observer reliability of the horse interactions confirms that the recorded interactions can easily be recorded.”
Line 401 suggest re-
R: We reworded it this way (line 411-412):
“This housing-related welfare impairment can lead to abnormal stereotypical behaviour, excessive aggressiveness towards humans [33], unresponsiveness towards their environment and an increase in alert posture [32].”
Line 416 suggest re-
R: This sentence was removed
Line 422
R: We reworded this slightly (line 430-435):
“In terms of learning theory, the driver and the groom added aversive stimuli using the rein, whip, voice, and hand signals in order to decrease the frequency of undesired behaviours, such as social interactions during work in pairs [34]. All stimuli were used in a fairly consistent manner: hand signals and pulling on the reins were most frequent, and of low intensity, while the whip was only used when the horse did not react to the reins or the voice.”
Line 433 suggest re-wording to
R: We reworded this section, and added some limitations as suggested by another reviewer (line 447-453).
“We limited the ethogram to behaviours that were obvious to the observer, with the lowest intensity of interaction consisting in one horse turning his head towards the other (“approach”). It is possible that more subtle clues between the two stallions might have been missed, and were not recorded for the analysis. However, the substantial intra-observer reliability shows that the behaviours we recorded could be reliably assessed from videos and indicate that the methodology is sound for further research. Furthermore, the substantial intra-observer reliability for the human interventions between live and video observations suggests that either direct recordings or assessing one video would be sufficient to record the total interactions between horses driven in pairs.”
Line 434 needs rewording.
R: See last comment.
Lines 435 438 needs rewording and the RM abbreviation is bolded when other abbreviations are not.
R: We reworded the last sentences to be more clear and removed the boldness (line 467-472).
Just a thought and I know it is not possible to collect new data on this, but it could be mentioned as a suggested future study: why not pair the horses driving who are housed together in the social boxes? As well as (or instead of) the neutral horses? Any info on dominance hierarchy between the horses?
R: We have added those limitations and suggestions as a final paragraph, thank you. (line 474-482)
Overall comment: this is a very interesting and valuable paper which is overall written very well. I think that you have an opportunity to present some detailed individual change data which may show strong effects of the social housing conditions as the mean data (which is so variable) may hide individual changes in interactions.
I enjoyed reading this, thank you.